# Update on Patient Self-Testing with Portable and Wearable Devices: Advantages and Limitations

**DOI:** 10.3390/diagnostics14182037

**Published:** 2024-09-13

**Authors:** Giuseppe Lippi, Laura Pighi, Camilla Mattiuzzi

**Affiliations:** 1Section of Clinical Biochemistry, University of Verona, P.le Ludovico Scuro 10, 37134 Verona, Italy; laura.pighi2@aovr.veneto.it; 2Medical Direction, Rovereto Hospital, Privincial Trust for Social and Sanitary Services, Corso Verona, 4, 38068 Rovereto, Italy; camilla.mattiuzzi@apss.tn.it

**Keywords:** laboratory medicine, wearable devices, portable devices, home testing, remote patient monitoring

## Abstract

Laboratory medicine has undergone a deep and multifaceted revolution in the course of human history, in both organizational and technical terms. Over the past century, there has been a growing recognition of the need to centralize numerous diagnostic activities, often similar or identical but located in different clinical departments, into a common environment (i.e., the medical laboratory service), followed by a progressive centralization of tests from smaller laboratories into larger diagnostic facilities. Nevertheless, the numerous technological advances that emerged at the beginning of the new millennium have helped to create a new testing culture characterized by a countervailing trend of decentralization of some tests closer to patients and caregivers. The forces that have driven this (centripetal) counter-revolution essentially include a few key concepts, namely “home testing”, “portable or even wearable devices” and “remote patient monitoring”. By their very nature, laboratory medical services and remote patient testing/monitoring are not contradictory, but may well coexist, with the choice of one or the other depending on the demographic and clinical characteristics of the patient, the type of analytical procedure and the logistics and local organization of the care system. Therefore, this article aims to provide a general overview of patient self-testing, with a particular focus on portable and wearable (including implantable) devices.

## 1. Introduction

Laboratory medicine has undergone a profound and multifaceted revolution in the course of human history, in both organizational and technical perspectives. Although it is impossible to pinpoint an exact date for the official “birth” of laboratory medicine, it is generally recognized that the introduction of modern clinical chemistry and microbiology in the early- to mid-19th century represented a milestone compared to previous decades [1]. From an organizational perspective, there was a growing recognition of the need to centralize in a common environment (i.e., the medical laboratory service) the numerous diagnostic activities that were often similar or identical but located in different clinical departments. This important revolution fostered many advantages resulting from the centralization of tests, such as lower costs due to the avoidance of redundancies, a high degree of standardization of analytical procedures and the development of a new profile of “laboratory professionals”, whose specific activity was now mainly focused on diagnostic tests [2]. In parallel with centralized testing, analytical techniques have evolved from rudimentary diagnostic methods to sophisticated technologies encompassing almost all areas of laboratory diagnostics (i.e., clinical chemistry, immunochemistry, hematology, hemostasis, molecular biology, etc.) that support modern healthcare [2]. Last but not least, advances in technology and informatics have paved the way for greater mechanization and automation, as well as for the ability to process a large amount of diagnostic information at lower cost. This has ultimately led to the emergence of the modern concept of laboratory medicine services, which are the ideal place where the vast majority of phenotypic and genotypic tests are performed in a modern healthcare facility [3].

While laboratory testing has gradually shifted from the bedside to the clinical laboratory over the last century, with sometimes excessive centralization leading to several million different tests being housed in the same (huge) diagnostic facility, the numerous technological advances at the beginning of the new millennium have helped to create a new testing culture, characterized by an opposite trend of decentralization of some tests closer to the patient and caregiver [4]. The forces that have driven this (centripetal) counter-revolution essentially include a few key concepts, namely “home testing”, “portable or even wearable devices” and “remote patient monitoring” [5]. Taken together, these concepts represent a paradigm shift in the way healthcare could be delivered, as this approach leverages technology to improve patient outcomes and increases convenience and often reduces the overall cost of healthcare compared to traditional testing performed in centralized laboratory services.

By their very nature, medical laboratory services and remote testing/monitoring of patients are not contradictory, but may well coexist, with the choice of one or the other depending on patient’s demographic and clinical characteristics, type of analytical procedure, logistics and local organization of the healthcare system [6]. Therefore, this article aims to provide a general overview of patient self-testing, with a particular focus on portable and/or wearable (including implantable) devices.

## 2. Current Wearable Medical Devices and Remote Patient Monitoring

A wearable medical device is usually defined as a tool that can be used by the patient to record data, usable for the care of the wearer [7]. Remote patient monitoring (often referred to as “telemonitoring”), on the other hand, is used to improve management using digitally transmitted health-related data [8]. The use of these devices for the remote monitoring of patients outside the specific area of in vitro diagnostics is now practically commonplace and has been hugely amplified by the recent coronavirus disease 2019 (COVID-19) pandemic, when direct contact between patients and clinicians was disrupted, paving the way for the use of new protocols and innovations to simultaneously ensure safety and maintain quality of care [9].

There are already many potential applications for wearable medical devices that can be used for real-time monitoring of a variety of physiological conditions, such as body temperature (wearable thermometers), blood pressure (blood pressure cuffs and monitors), heart rhythm (holters, wearable electrocardiography (ECG) devices), oxygen saturation (pulse oximeters), blood perfusion of dermis and subcutaneous tissue (wearable photoplethysmogram (PPG) devices), brain activity (wearable electroencephalogram (EEG) devices), muscle response to nerve stimulation (wearable electromyogram (EMG) devices), driving eye movements (wearable electrooculography (EOG) devices) and vocal fold vibration during voice production (wearable electroglottogram (EGG)) [10,11], but also artificial intelligence (AI)-based cough sound detection devices developed to identify specific variations in cough sounds that may be frequent in patients with COVID-19 [12]. The following part of this article instead provides a general overview of the established devices, followed by some innovative portable or wearable devices developed for the self-testing of some important laboratory parameters.

## 3. Some Paradigmatic Examples of Well-Established Portable Laboratory Testing Devices

In addition to the conventional wearable or portable devices that have been developed to monitor physiological functions, there are other instruments that allow patients to monitor and measure various analytes, which can then be used to diagnose and, in particular, monitor pathologies The potential advantages of these techniques are manifold and include time savings (i.e., avoidance of phlebotomy in specialized facilities), less invasive means of analytes assessment, real-time measurement of the target analyte, easy collection of test results, possibility of (remote) health monitoring and patient empowerment. All of these potential benefits are leading to a relentless deployment of these devices on the diagnostic market, which we aim to briefly update here. For this purpose, we have arbitrarily selected two paradigmatic examples (i.e., monitoring of oral anticoagulant therapy and glucose) to provide a general introduction to the problem in relation to two of the most commonly used approached currently available for patient monitoring.

### 3.1. Monitoring of Oral Anticoagulant Therapy

In patients requiring anticoagulant medication, drug monitoring is an important support to reduce the risk of occurrence or recurrence of thrombosis [13]. For decades, anticoagulant therapy has been based on the so-called vitamin K antagonists (VKAs), namely warfarin and acenocoumarol, whose dynamic activity in the blood has a narrow range, so that constant monitoring is required to prevent under- or over-coagulation [14]. For many years since their introduction, VKAs were monitored in clinical laboratories using the prothrombin time/international normalized ratio (PT/INR), which was inconvenient for patients as they had to have blood drawn regularly at phlebotomy centers and wait for test results for therapeutic adjustments. The advent of new drugs such as direct oral anticoagulants (DOACs), which have replaced VKAs in many clinical applications, has certainly reduced the need for routine blood testing, as these innovative agents require less stringent measurement of their activity in plasma [15]. Nevertheless, VKAs remain the first choice for some clinical indications, and their use is still preferred in some countries and/or also in certain patients in whom DOACs may be discouraged and/or regular INR measurement would more reliably reflect the anticoagulant effect [15]. 

The need to frequently test patients receiving VKA therapy has led the diagnostics industry to develop portable devices that allow for near-patient testing, commonly referred to as portable coagulometers. Specifically, regarding the use of these patient-friendly devices in conjunction with telemedicine for anticoagulant dose adjustment, a recent meta-analysis published by Huang et al. showed that this approach can reduce testing frequency by more than 12 days, significantly improves time spent in the therapeutic range by nearly 10% and reduces the risk of venous thromboembolism by nearly 30%, while leaving the risk of major bleeding, re-hospitalization and mortality unchanged [16]. 

In light of these findings, there are now innovative approaches to monitoring anticoagulant therapy without the need to use portable coagulometers. For example, Chan et al. have developed a revolutionary INR testing system based on the vibration motor and camera of a regular smartphone [17]. In short, capillary blood is deposited in a plastic cup containing tissue factor and a copper particle. When the smartphone vibration is activated, the continuous vibration of the copper particle is monitored by the smartphone’s camera. When hemostasis is activated by tissue factor, a clot starts to form in the cup and traps the copper particle, whose movement gradually slows down until it reaches a stationary phase, which defines the clotting time value. The correlation of this INR measurement with that performed in the same cohort of patients using a standard coagulometer was 0.97. 

In another article, Xu et al. developed another portable, smartphone-based device to determine INR in whole blood [18]. In brief, a small amount of sample is collected in a plastic cup, mixed with tissue factor, and placed in a microfluidic device for flow rate measurement, which is equipped with light-emitting diodes (LEDs) on the side wall that can be operated with different types of smartphone. Immediately afterwards, the smartphone’s video recorder is activated to record the image of the sample flow in the microfluidic channel. When the solution in the chamber reaches a steady state, the video is stopped and the total time elapsed between sample flowing and steady state is recorded (basically, the number of pixels in the channel of the microfluidic device increases as soon as flow starts and stops when a clot forms). The correlation (square r; R^2^) with the INR value of plasma samples measured with a conventional coagulometer was 0.90.

### 3.2. Blood Glucose Monitoring

According to the American Diabetes Association (ADA), blood glucose monitoring (BGM) is an important pillar in the management of patients with diabetes [19]. Regardless of the importance of BGM for detecting hypoglycemia (e.g., at fasting, during bedtime, during and after exercise), suspected hyperglycemia and before or during critical tasks (e.g., driving a car), the ADA provides additional recommendations on this topic, i.e., continuous glucose monitoring (CGM) must be made available to all patients with type 1 diabetes as soon as possible (preferably immediately after diagnosis). The type of device must be personalized according to the patient’s specific needs, preferences and abilities, and initial and ongoing education and training must be provided to all patients and/or caregivers [19].

CGM is increasingly used in insulin-dependent diabetics to better control their blood glucose fluctuations, thus enabling a better management of diabetes by adjusting lifestyle or insulin in real time [20]. These devices can be considered the archetype of wearable devices as they usually consist of three parts, i.e., (i) a small electrode placed under the skin (usually on the abdomen or arms) with an adhesive patch to measure the glucose content in the interstitial fluid using enzymatic technology; an electric current proportional to the analyte concentration is generated, which is then transmitted to (ii) a transmitter that communicates the electrode readings at regular intervals (e.g., between 1 and 15 min) to (iii) a software program located in a “receiver”, which ultimately displays the glucose level. The receiver can be a special device, a smartphone, a smartwatch or even an insulin pump (Figure 1). 

There are three types of CGM device, differentiated by the way the information is transmitted and displayed. The first type (“real-time”) of CGM device automatically sends the information to the software at a specific time, a second type (“intermittent-scan”) also measures glucose levels continuously, but the skin patch must be scanned with the receiver at specific times to display and store the information, while a third type of devices only collects data that can be downloaded and reviewed at a later stage [20]. The biggest advantage of the latest generation of CGM is that they can display blood glucose levels in real time as a graph and provide alerts when the glucose levels drop or rise rapidly, thus reducing the risk of hypo- or hyperglycemia [21]. 

This first clinical application of CGM was in type 1 diabetics. The advantages of this approach compared to standard CGM were highlighted in the meta-analysis by Teo et al. (including 21 studies and 2149 participants) [22]. In brief, the use of CGM was associated with a significant reduction in glycosylated hemoglobin (HbA1c) levels compared with standard BGM (mean difference: −2.46 mmol/mol), with greater effects in participants with higher baseline HbA1c (i.e., >64 mmol/mol; mean difference: −4.67 mmol/mol). A trend towards a lower risk of severe hypoglycemia was also seen in patients using CGM (−39%; *p* = 0.13). A recent meta-analysis published by Jancev et al. (including 12 randomized controlled trials and 1248 participants) found that CGM is also preferable to BGM in type 2 diabetics [23]. The authors found that the use of CGM was associated with a lower value of HbA1c (−0.31%), longer time in the normal range (+64%), shorter time below (−0.66%) and above (−5.86%) the normal glucose range and lower glycemic variability (−1.47%) compared to standard BGM.

In addition to the traditional transdermal sensors for CGM, there are new emerging alternatives. Chang et al. developed an integrated smartwatch for non-invasive CGM [24]. The device is based on a nafion-coated flexible electrochemical sensor patch integrated into the wristband to allow transdermal contact with the interstitial fluid via iontophoresis-based extraction, measuring glucose with glucose oxidase by assessing the amperometric response triggered by the hydrogen peroxide (H_2_O_2_) generated when reacting with glucose. All electrical parts are included in the watch, including a rechargeable power source and modules for signal processing and wireless transmission. The glucose values are then displayed in real time on the LED screen. A preliminary test including a cohort of 23 volunteers and using a Clarke error grid showed that this innovative device has an accuracy of 84.3% in measuring glucose levels compared to the value obtained with a commercial portable glucometer [24].

## 4. Innovative Wearable Laboratory Testing Devices

Along with the traditional wearable or home testing devices developed for monitoring major diseases such as diabetes and anticoagulant therapy, other instruments are now entering the market that enable patient self-testing based on transdermal biosensors, which allow the measurement of a variety of other analytes that diffuse from the bloodstream through the skin and can thus be used for the diagnosis and monitoring of major human diseases [25]. Although we will not provide a comprehensive description of all potential applications of wearable devices for measuring laboratory biomarkers (e.g., hemoglobin, lactate, bilirubin, creatinine, ions, alcohol, hormones, among others), following is a brief explanation of the three leading technologies (e.g., microfluidics, microneedling, reverse iontophoresis). Firstly [26], microfluidic sensors usually consist of four parts (microchannels for transporting the biofluid through the sensor, microvalves for directing the fluid, micropumps to deliver the fluid into the device and the sensor where the measurement takes place). These devices are designed to use very small volumes of biofluids (e.g., interstitial fluid and sweat) for real-time monitoring of the target analyte. Microneedles, which can be made of various materials such as silicon, polymers or hydrogels, penetrate the outer layer of the skin to gain direct access to the interstitial fluid, which is then transferred to specific optical or electrochemical sensors for measuring the target analyte. Reverse iontophoresis is the inverse principle of iontophoresis, as the system is designed for the extraction of molecules rather than delivery. A weak electrical current is applied across the skin electrodes, creating an electrical gradient that draws ions and other molecules from the interstitial fluid to the skin surface where they can be analyzed by specific sensors.

The following part of this article summarizes some paradigmatic examples covering two very common pathologies (myocardial infarction and sepsis) where early diagnosis is one of the most important factors for influencing the prognosis.

### 4.1. Cardiac Troponin Testing

Cardiac troponin testing, which includes either cardiac troponin I (cTnI) or cardiac troponin T (cTnT), is the most important basis for the diagnosis of acute coronary syndrome (ACS), as all developed guidelines now include the assessment of these biomarkers in diagnostic algorithms [27,28]. The availability of accurate and timely cardiac troponin measurements is therefore crucial for reducing the risk of adverse outcome in patients with myocardial ischemia, as every minute of delay counts. To this end, a seminal article by De Luca et al. [29] showed that each 30-minute delay in the diagnosis of acute ST-segment elevation myocardial infarction (STEMI) was associated with a 7% higher risk of 1-year mortality (relative risk: 1.07; 95%CI, 1.01–1.15).

It is therefore not surprising that a variety of point-of-care tests (POCT) and rapid diagnostic tests have been developed and marketed to meet the increasing demand for the early diagnosis and treatment of ACS [4,30]. In this context, some interesting alternatives have also emerged. Titus et al. constructed a wrist-worn transdermal infrared spectrophotometer sensor to measure cTnI within 5 min and without sampling [31], which was subsequently tested in a cohort of 134 patients diagnosed with ACS [32]. In a subsequent validation cohort consisting of 45 patients with suspected ACS, the comparison of results of the device with those of traditional laboratory testing yielded an accuracy (area under the curve, AUC) of 0.92 (95%CI, 0.80–0.98), combined with a sensitivity of 0.94 and a specificity of 0.64, demonstrating that infrared spectroscopic measurement of cTnI through the skin is a viable and particularly accurate option for self-patient testing in those at increased risk of myocardial ischemia. 

### 4.2. Sespsis Diangosis

Like ACS, sepsis is a life-threatening disease that must be diagnosed as soon as possible, as synthesized in the acronym TIME (“Temperature”, “Infectious symptoms”, “Mental decline”, “being Extremely ill”) [33]. The imperative need for early diagnosis and timely treatment of sepsis has been widely emphasized and is also a feature of the Third International Consensus Definitions for sepsis and septic shock [34]. To date, a variety of biomarkers have been proposed to enable a more rapid diagnosis of sepsis (e.g., C-reactive protein, procalcitonin, presepsin), the measurement of which is now also possible with POCT devices [35]. Nevertheless, some interesting alternatives are also emerging in this case.

Li et al. developed a soft, portable and battery-free wound dressing system for real-time monitoring of wound condition and measurement of procalcitonin in wound exudate [36]. Once applied to a wound and connected to a portable detector (i.e., a smartphone) used for wireless power, data processing and transmission, the device enables simultaneous measurement of temperature, pH and procalcitonin. In a mouse model of lipopolysaccharide-induced sepsis, the device was able to detect increased procalcitonin levels in the exudate just 2 h after the injection of lipopolysaccharides. It is therefore conceivable that the use of this wound dressing system for continuous monitoring of procalcitonin levels in patients with an increased risk of sepsis will enable very early diagnosis of sepsis and can also be used to monitor antimicrobial therapy to enable timely reduction or discontinuation of antibiotics as soon as procalcitonin levels change [37].

## 5. Potential Problems of Portable or Wearable Lab Testing Devices

According to a general definition, POCT can be divided into near-patient testing (NPT) and patient self-testing (PST). While NPT is performed and clinically interpreted by healthcare professionals, PST is performed and interpreted directly by laypersons, with all the associated risks [38]. Therefore, the proliferation of wearable devices that enable PST would raise important technical, clinical and ethical issues, the understanding of which is essential before this new technology would be ready for widespread usage. There are several potential problems associated with PST, which are directly related to the brain-to-brain loop that characterizes the total testing process [39], and therefore may be related to the pre-analytical, analytical and post-analytical phases, as summarized in Table 1.

### 5.1. Pre-Analytical Issues

Wearable medical devices can basically be divided into those that are physically connected to the skin and directly measure analytes in the interstitial fluid (e.g., CGM sensors) and those that are simply worn without any direct fluid exchange between the device and the skin. Regardless of the type of device, they ultimately raise a number of potential drawbacks. Although regulations for the procurement of PST devices are extremely heterogeneous around the world, some of them can be easily purchased over the internet, paving the way for significant risks. The first is the cost to patients and to the healthcare system as a whole. If not directly covered by the healthcare system or insurance, the cost of some of these devices may be prohibitive for certain categories of patients. Therefore, an appropriate health technology assessment (HTA) must be conducted prior to the introduction of innovative devices to measure certain analytes for the diagnosis/monitoring of patients, the costs of which are ultimately borne by the healthcare system [40]. Another critical aspect is that some devices may be sold without official approval from regulatory authorities such as the FDA in the US or the European Community (CE) in Europe, they may contain falsified clearance or approval claims, they may not be the most appropriate type of device for certain types of patients, their use may be restricted to healthcare professionals or, worse still, they may be counterfeit [41].

The second important aspect is that the device may not be appropriate for the patient’s condition, as it could be used by healthy patients, thus increasing the risk of false positive readings and/or overdiagnosis. Another source of problems is the way the device (or its sensor) is placed. Whether the device is “implanted” or more generally “worn”, it must be positioned correctly, avoiding displacement, in order to function appropriately and stably. If it is a sensor, the area of skin to which the device is attached must be adequately prepared according to the specific instructions provided by the manufacturer [42]. Patient status must also be considered when interpreting test results. For example, hydration status, body temperature, peripheral blood perfusion, hematocrit, partial oxygen pressure (pO_2_), fasting status and even bilirubin and urate can affect the accuracy of CGM readings [43,44]. The long-term complications of wearing implantable devices must also be considered. A recent meta-analysis of continuous or flash glucose monitoring devices concluded that the cumulative rate of skin complications was one episode every eight weeks of wear, of which 1.5% were classified as serious [45]. The most frequent cutaneous complication was irritation/erythema (55% of all complications), followed by pruritus/itching (11%), induration (8%) and edema (7%). Pain upon sensor insertion and mild bleeding have also been occasionally documented. Some disposable devices should therefore be replaced regularly, depending on the model. If this is neglected or forgotten, the patient may be exposed to an increased risk of malfunction when the device reaches the end of its service life.

### 5.2. Analytical Issues

Although laboratory errors in the analytical phase are less common compared to mistakes in the other phases of the total testing process, the problem is magnified to some extent with portable (and wearable) testing devices, as these tools are intended for self-use directly by the patients, with little/no oversight by laboratory personnel. In the vast majority of cases, patients have little background knowledge of the technical function of these devices, calibration, quality control and maintenance. The probability of a portable/wearable device not performing its measurement correctly is therefore hypothetically higher than for a conventional test in a medical laboratory service. To this end, specific requirements are needed to assess the accuracy of these innovative devices, which could follow the performance specification-based approach already established for CGM [46]. Most of these devices, especially those used for CGM, need to be regularly calibrated with blood measurements to provide accurate readings, while inaccurate calibration may lead to significant discrepancies between CGM readings and actual blood values, which may ultimately affect patient safety. For all potential devices that will become commercially available, pre-market validation studies conducted by recognized scientific organizations or renowned scientists and regular comparison of analytical values with test results obtained from patient samples using laboratory-based methods (e.g., during health check-ups) would be essential to ensure the accuracy and reliability of the data collected by wearable devices. Another important aspect that patients should be made aware of is the interference with some substances, which can lead to falsely elevated or lowered values of the measured analyte. Chemical interference can be caused, for example, by the presence of paracetamol, hydroxyurea, ascorbic acid, dopamine, mannitol or icodextrin [47].

Environmental conditions are other factors that can affect the actual readings of these portable/wearable devices, as the readings of the measured analytes can be influenced in particular by extreme temperatures, humidity or altitude, as well as possible interference from sweat, body lotions applied to the skin, physical pressure on the sensor and electromagnetic radiation [44]. High temperatures can accelerate enzyme activity in the sensor, leading to falsely elevated values, or accelerate sensor degradation, while cold temperatures can shorten the response time. High humidity can also affect the adhesive properties of the sensor patch and potentially affect its correct placement. Altitude differences, especially high altitudes, reduce oxygen levels and thus alter the readings of sensors that utilize some enzymatic reactions [43,44]. Establishing valid quality control is therefore crucial to ensure the accuracy of the readings and also to minimize the potential batch-to-batch variations of the sensors and for detecting potential sensor deviations in a timely manner [48].

### 5.3. Post-Analytical Issues

The frequently neglected or underestimated risk of the post-analytical phase, i.e., the improper recording and interpretation of laboratory data, should not be neglected and is certainly exacerbated in the context of patient self-testing [49]. This is mainly due to two aspects, namely problems in the transmission of information from patients to clinicians or nursing staff and self-interpretation of test results. With the increasing connectivity of medical devices, of which wearable devices are a paradigmatic example, the risk of cyberterrorism increases exponentially through infection with malware or other software programs that can capture or corrupt patient data and cause device malfunction [50]. In particular, most wearable medical devices use patient accounts that may contain sensitive information such as health records and other personal details (e.g., social security numbers, credit card numbers), so that data breaches can also be used to demand ransom. 

Lack of connectivity due to device or network malfunctions can result in the disruption of transmission of test results to the personnel responsible for interpreting the information and taking medical decisions, and is hence another tangible risk of portable/wearable laboratory testing devices. Transcription errors or incorrect data entry are other errors that can occur in the post-analytical phase if the portable/wearable devices are not (or cannot be) directly connected to the medical record [51]. 

Errors in data interpretation are another important source of post-analytical mistakes, as misinterpretation of test results following self-testing can lead to incorrect therapeutic decisions or inappropriate adjustment of therapies [52]. Some of these devices are capable of generating real-time data monitoring of some analytes (e.g., CGM), and this large volume of information may be overwhelming as it can be challenging for the patient to analyze and respond to such a large amount of data provided in a very short time. Full integration of the data generated by wearable laboratory testing devices is then required, as this information not only needs to be permanently stored in the patient history, but also interpreted in the context of other clinical and laboratory information [53]. The variability of reporting formats and reference ranges can then contribute to complicating the interpretation of readings, especially when integrating data from different sources [49].

## 6. All That Glitters Is Not Gold

In addition to wearable devices that are capable of assessing some analytes by direct measurement, there is a plethora of new smartwatches that are increasingly being marketed as providing data on a variety of blood parameters. The problem is that these devices do not measure the analyte directly but estimate its concentration through a combination of physical measurements (e.g., photoplethysmography, electrocardiography, accelerometry, bioelectrical impedance analysis, skin temperature monitoring) integrated with machine-learning algorithms. Thus, the final value does not reflect the actual (measured) concentration of the analyte but is rather an estimated value based on various physical characteristics of the patient. The performance of one such system has been tested in a study conducted by Dun et al. [54]. Information on skin temperature, heart rate, physical activity and electrodermal activity was combined with statistical learning models to predict the concentration of 44 analytes of clinical utility in primary care (e.g., complete blood count and differential, metabolic, liver and kidney panels). The comparison (i.e., the correlation coefficient) with conventional laboratory assays was always less than 0.5, with only two measures (hemoglobin and red blood cells) showing a correlation greater than 0.4. Combining values of tests within pre-defined panels yielded even lower cumulative correlations (i.e., always <0.4), with over 75% of measures yielding a correlation ≤0.1.

There are currently a number of commercially available smartwatches equipped with laboratory parameter estimation features, some of which are in the early stages of deployment. They are all basically similar in that they track a range of physiological parameters and estimate the values of a range of blood analytes through back-end calculations. Nevertheless, manufacturers are required to validate analytical performance in independent studies before these devices can be considered as potential replacements for the direct determination of analytes.

## 7. Conclusions

Portable and especially wearable laboratory testing devices are already a reality in certain clinical situations (such as GCM) and will soon be a major opportunity for the future of laboratory medicine, as CGM test results were found to be highly comparable to those measured on whole blood with conventional techniques [55,56,57]. The use of portable laboratory testing devices offers several potential advantages, such as the fact that the blood does not have to be drawn directly at phlebotomy centers, the reduced anxiety before receiving test results from traditional medical laboratories, the relatively less invasiveness of the latest generation of these devices compared to a standard venipuncture, the possibility of real-time monitoring of analytical values (and their fluctuations) with timely clinical response to abnormal readings, the wireless connection to medical records allowing remote monitoring of the patient by physicians and/or care givers and, last but not least, greater patient empowerment.

Some intriguing opportunities are also emerging. The universal use of smartphones is paving the way for their use as diagnostic devices to capture, interpret or transmit test results when directly connected to a lateral flow immunoassay. This is mainly due to the high-performance cameras that modern smartphones are equipped with, which may serve as sensors for converting images into electrical signals that would allow semi-quantitative or even quantitative assessment of test results. Several applications have already been proposed, as summarized in the recent review article by Park [58]. Although we have recognized the numerous possibilities and potential clinical benefits of these innovative devices, some important drawbacks, summarized in Table 1, must be considered before the use of these devices becomes widespread. There are also some potential solutions that could mitigate the current limitations of wearable laboratory testing devices, mostly encompassing higher efficiency and better accuracy by adopting advanced physical techniques, improved (e.g., miniaturization) sensor design that could reduce potential discomfort, improved data management by wireless connectivity and, last but not least, integration of artificial intelligence algorithms that may improve data interpretation and foster predictive analytics [59,60].

## Figures and Tables

**Figure 1 diagnostics-14-02037-f001:**
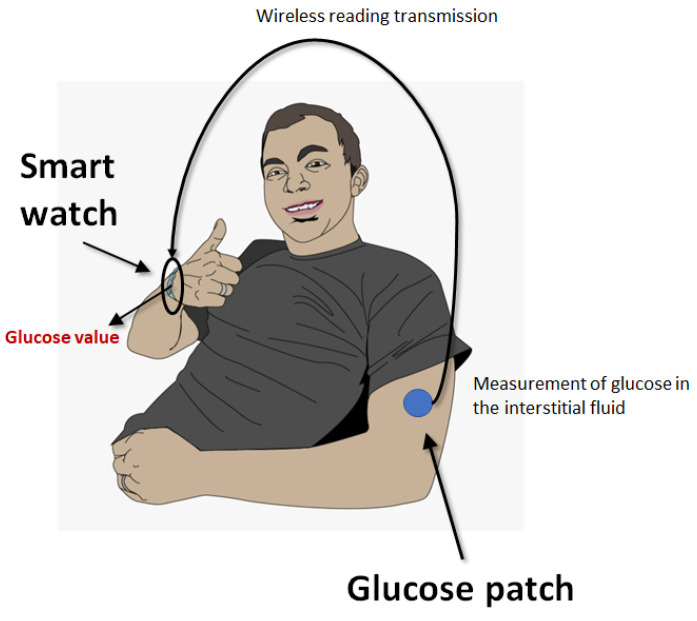
Example of continuous glucose monitoring, composed of a skin patch containing an electrode for measuring glucose content in the interstitial fluid, which is then transmitted with wireless connection to a wearable device (i.e., a smartphone), where patients can read their actual glucose concentration.

**Table 1 diagnostics-14-02037-t001:** Pre-analytical, analytical and post-analytical problems of portable or wearable lab testing devices.

Phase of the Testing Process	Problem
Pre-analytical	Regulatory challenges
	Cost
	Inappropriate purchasing
	Patient variables
	Appropriateness
	Sensor placement
	Maintenance and replacement
	Long-term injuries
Analytical	Calibration
	Chemical interreference
	Environmental conditions
	Lot-to-lot variation
	Cyberterrorism
	Lack of connectivity
	Transcription errors
Post-analytical	Cyberterrorism
	Lack of connectivity
	Transcription errors
	Misinterpretation of test results
	Information overload
	Integration within the medical record
	Variability in reporting formats and reference ranges

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
