# Peer review of "Update on Patient Self-Testing with Portable and Wearable Devices: Advantages and Limitations"

_diagnostics, 2024, doi:10.3390/diagnostics14182037_

Round 1

Reviewer 1 Report

Comments and Suggestions for Authors

This perspective paper provides an overview of wearable/portable devices in patient self-testing. It is very well-written and the reviewer suggests the manuscript be accepted after addressing the following minor recommendations.

1.      Since there are lots of examples of wearable/portable medical devices for measuring different other vital parameters such as blood pressure, why do the authors specifically focus on monitoring oral anticoagulant therapy and blood glucose?

2.      The reviewer suggests that the authors can provide some schematics or images from the cited paper to better illustrate the working principles of the reported medical devices in Section 3.

3.      When introducing the reported medical devices, the reviewer recommends that the author can elaborate more on the principles in Section 4.

4.      In Section 5, it is advisable for the authors to provide some potential solution that can mitigate the current limitations to further enhance the impact of this manuscript. 

Author Response

  1. Since there are lots of examples of wearable/portable medical devices for measuring different other vital parameters such as blood pressure, why do the authors specifically focus on monitoring oral anticoagulant therapy and blood glucose?
  • ANSWER: Thanks for this comment. As mentioned, “We have arbitrarily selected two paradigmatic examples (i.e., monitoring of oral anticoagulant therapy and glucose) to give a general introduction to the problem in relation to two of the most commonly used approached available currently available for patient monitoring”. This explanation has been included in the text of the article. Certainly, other examples could be done, but we have been more familiar with these two and we felt more “secure” to discuss them.

  1. The reviewer suggests that the authors can provide some schematics or images from the cited paper to better illustrate the working principles of the reported medical devices in Section 3.
  • ANSWER: Thanks for this comment, we found it very useful. We have included an image on how a glucose test patch works, as a reliable example of some of these techniques (new Figure 1).

  1. When introducing the reported medical devices, the reviewer recommends that the author can elaborate more on the principles in Section 4.
  • ANSWER: Thanks for this comment. This is very good suggestion. We have hence expanded this part, as follows: “In brief [26], microfluidic sensors usually consist of four parts (microchannels for transporting the biofluid through the sensor, microvalves for directing the fluid, micropumps to deliver the fluid into the device, and the sensor where the measurement takes place). These devices are designed to use very little volumes of biofluids (e.g., interstitial fluid and sweat), for real-time monitoring of the target analyte. Microneedles, which can be made of various materials such as silicon, polymers or hydrogels, penetrate the outer layer of the skin to gain direct access to the interstitial fluid, which is then transferred to specific optical or electrochemical sensors for measuring the target analyte. Reverse iontophoresis is the inverse principle of iontophoresis, as the system is designed for extraction of molecules rather than delivery. A weak electrical current is applied across the skin electrodes, creating an electrical gradient that draws ions and other molecules from the interstitial fluid to the skin surface where they can be analyzed by specific sensors”.

  1. In Section 5, it is advisable for the authors to provide some potential solution that can mitigate the current limitations to further enhance the impact of this manuscript.
  • ANSWER: Thanks for this comment. We have substantially modified the manuscript also addressing the comments of the second referee and these changes have also involved some consideration to mitigate the current limitations. We have however also include a final remark, following the suggestion of the referee, as follows: “There are also some potential solutions that could mitigate the current limitations of wearable laboratory testing devices, mostly encompassing higher efficiency and better accuracy by adopting advanced physical techniques, improved (e.g., miniaturization) sensor design that could reduce potential discomfort, improved data management by wireless connectivity and, last but not least, integration of artificial intelligence algorithms that may improve data interpretation and foster predictive analytics [62,63].”

Reviewer 2 Report

Comments and Suggestions for Authors

The article focus on patient self-testing and wearable devices is timely and relevant given the increasing emphasis on personalized healthcare and patient empowerment.

Title is too generic, suggested to change the title.

There is no clarity in the problem statement , authors must clearly state the problem and significance of solution.

Authors must add in existing literatures and highlight the gaps and provide reasons for developing portable devices.

Author must address how to ensure the accuracy and reliability of data collected from wearable devices?

The work can be compared with traditional testing methods.

Article must include potential risks and benefits of using wearable devices for health monitoring.

There will be regulatory challenges associated with the development and use of wearable devices, it can be included in future works.

Comments on the Quality of English Language

Grammar check is required

Author Response

Title is too generic, suggested to change the title.

  • ANSWER: Thanks for this comment. We agree. The title has been changed as follows: “Update on patient self-testing with portable and wearables de-vices: advantages and limitations”

There is no clarity in the problem statement, authors must clearly state the problem and significance of solution.

  • ANSWER: Thanks for this comment. Done, as follows: “The potential advantages of these techniques are manifold and include time savings (i.e., avoidance of phlebotomy in specialized facilities), less invasive means of analytes assessment, real-time measurement of the target analyte, easy collection of test results, possibility of (remote) health monitoring and patient empowerment. All of these potential benefits are leading to a relentless deploy of these devices in the diagnostic market, which we aim to briefly update here.”. The final part of the article has also been addressed accordingly. See answer to referee 1.

Authors must add in existing literatures and highlight the gaps and provide reasons for developing portable devices.

  • ANSWER: Thanks for this comment. Done, as follows: “The potential advantages of these techniques are manifold and include time savings (i.e., avoidance of phlebotomy in specialized facilities), less invasive means of analytes assessment, real-time measurement of the target analyte, easy collection of test results, possibility of (remote) health monitoring and patient empowerment. All of these potential benefits are leading to a relentless deploy of these devices in the diagnostic market, which we aim to briefly update here.”.

Author must address how to ensure the accuracy and reliability of data collected from wearable devices?

  • ANSWER: Thanks for this comment. This is a good point. We have discussed this aspect as follows: “For all potential devices that will become commercially available, pre-market validation studies conducted by recognized scientific organizations or renowned scientists and regular comparison of analytical values with test results obtained from patient samples using laboratory-based methods (e.g. during health check-ups) would be essential to ensure the accuracy and reliability of data collected by wearable devices”

The work can be compared with traditional testing methods.

  • ANSWER: Thanks for this comment. We have included that, when available (for sepsis no comparison was made, but for the troponin device this information was captured and now included, as follows: “In a subsequent validation cohort consisting of 45 patients with suspected ACS, the comparison of results of the device with those of traditional laboratory testing yielded an accuracy (area under the curve; AUC) of 0.92 (95%CI, 0.80-0.98), combined with a sensitivity of 0.94 and a specificity of 0.64”). For CGM, which is already a consolidated technique suggested also by the ADA, we just cited added pertinent references [3, in particular], as the good correlation with blood glucose is given already for granted and the readers can easily access those articles.

Article must include potential risks and benefits of using wearable devices for health monitoring.

  • ANSWER: Thanks for this comment. Done, as follows: Benefits: “The use of portable laboratory testing devices offers several potential advantages, such as the fact that the blood does not have to be drawn directly at phlebotomy centers, the reduced anxiety before receiving test results from traditional medical laboratories, the relatively less invasiveness of the latest generation of these devices compared to a standard venipuncture, the possibility of real-time monitoring of analytical values (and their fluctuations) with timely clinical response to abnormal readings, the wireless connection to medical records allowing remote monitoring of the patient by physicians and/or care givers and last but not least, the greater patient empowerment.”. Risks: “A recent meta-analysis of continuous or flash glucose monitoring devices concluded that the cumulative rate of skin complications was one episode every eight weeks of wear, of which 1.5% were classified as serious [45]. The most frequent cutaneous complication was irritation/erythema (55% of all complications), followed by pruritus/itching (11%), induration (8%) and edema (7%). Pain upon insertion sensor insertion and mild bleeding have also been occasionally documented.”. Then, there are the risk connected to inaccurate reading, which have already been discussed in the “post-analytical” section.

There will be regulatory challenges associated with the development and use of wearable devices, it can be included in future works.

  • ANSWER: Thanks for this comment, good point. We have included this point at the first place in table 1!

Round 2

Reviewer 2 Report

Comments and Suggestions for Authors

Revisions are statisfactory, It can be accepted